# Value of Left Ventricular Indexed Ejection Time to Characterize the Severity of Aortic Stenosis

**DOI:** 10.3390/jcm11071877

**Published:** 2022-03-28

**Authors:** Gabriele Pestelli, Valeria Pergola, Giuseppe Totaro, Marco Previtero, Patrizia Aruta, Antonella Cecchetto, Andrea Fiorencis, Chiara Palermo, Sabino Iliceto, Donato Mele

**Affiliations:** 1Cardiology Unit, Morgagni-Pierantoni Hospital, 47121 Forli, Italy; gabri.pestelli@gmail.com; 2Cardiovascular Research Unit, Fondazione Sacco, 47121 Forli, Italy; 3Department of Cardiac Thoracic Vascular Sciences and Public Health, University of Padova Medical School, 35128 Padova, Italy; valeria.pergola@gmail.com (V.P.); giustot88@gmail.com (G.T.); marco.previtero@gmail.com (M.P.); patrizia.aruta@aopd.veneto.it (P.A.); antonella.cecchetto@aopd.veneto.it (A.C.); andrea.fiorencis@gmail.com (A.F.); chiara.palermo@unipd.it (C.P.); sabino.iliceto@unipd.it (S.I.)

**Keywords:** aortic stenosis, echocardiography, ejection time, stroke volume, transvalvular flow rate, low-flow low-gradient aortic stenosis, hemodynamic

## Abstract

Aims: The assessment of aortic stenosis (AS) severity is still challenging, especially in abnormal hemodynamic conditions. Left ventricular ejection time (LVET) has been historically related to AS severity, but it also depends on heart rate (HR) and systolic function. Our aim was to verify if correcting LVET (LVET index, LVETI) by its determinants is helpful for the assessment of AS severity, irrespective of hemodynamic conditions. Methods and results: We retrospectively studied 152 patients with AS and 378 patients with heart failure and no-AS. At multivariate analysis, LVET (assessed with pulsed-wave Doppler) showed a strong correlation with stroke volume index (SVI) (Beta 0.354; *p* < 0.001), HR (−0.385; *p* < 0.001), AS grade (Beta 0.301; *p* < 0.001) and, less significantly, ejection fraction (LVEF) (Beta 0.108; *p* = 0.001). AS grade was confirmed to be a major determinant of LVET, irrespective of forward flow (assessed by SVI and transvalvular flow rate) and LVEF (above and below 50%). A regression equation was derived to index LVET (LVETI) to HR and SVI. By using this formula, LVETI detected severe AS more accurately (AUC 0.812, *p* < 0.001) than LVET alone (AUC 0.755, *p* for difference = 0.005). Similar results were observed in patients with abnormal flow status. As an exploratory finding, we observed that the highest tertile of LVETI was associated with a higher rate of aortic valve interventions during follow-up. Conclusions: LVETI correlates with AS severity better than uncorrected LVET, independently from hemodynamic conditions, and may help to discriminate severe AS. This finding needs confirmation in larger prospective multicenter studies.

## 1. Introduction

Severe aortic stenosis (SAS) is defined, according to recent echocardiographic guidelines, by a functional aortic valve area (AVA) ≤ 1.0 cm^2^, an indexed AVA ≤ 0.6 cm^2^/m^2^, a transvalvular mean pressure gradient (MG) ≥ 40 mmHg and/or a peak jet velocity (V_max_) ≥ 4 m/s [1]. Unfortunately, discrepancies between different measures of AS severity are encountered in clinical practice, making the diagnosis of SAS a challenging task. A reduced transvalvular forward flow is considered the main cause of discrepancy, in addition to technical errors [2,3]. Thus, other measures of AS severity are needed to echocardiographically allow the identification of SAS. Ideally, these measures should be simple, independent of heart rate (HR) and left ventricular (LV) systolic function, and should be applicable in different forward flow conditions.

LV ejection time (LVET) is a potential measure of SA severity. It draws inspiration from the semeiological analysis of the arterial systolic pulse in patients with AS, characterized by prolongation of the upstroke phase with decreased peak (pulsus parvus et tardus) in association with an overall pulse prolongation. With the spreading of Doppler-echocardiography, LVET was measured on a transaortic CW Doppler as the time interval from AV opening to AV closure [4], although other methods were also tested (e.g., LV outflow tract (LVOT) pulsed-wave Doppler, Doppler tissue imaging velocity curves) [5].

LVET is known to be prolonged in AS and it is included in the Gorlin formula for AVA calculation [6]. However, it is also affected by HR and LV systolic function, and this might preclude its use for the recognition of SAS patients. Other determinants could also influence LVET. Whether LVET, corrected for its determinants (indexed LVET, LVETI), correlates with AS severity in different forward flow conditions has never been demonstrated.

The primary objective of this study was to verify the impact of LVETI on AS grading, even in different forward flow conditions. Then, as secondary and exploratory objectives, we sought to verify whether LVETI is associated with surgical or percutaneous aortic valve intervention at follow-up. Such information could be helpful to set up a large-scale validation study of LVETI as a predictor of the severity of AS.

## 2. Materials and Methods

### 2.1. Patient Groups

Echocardiograms from 233 consecutive adult patients with new or confirmed diagnosis of moderate-to-severe valvular AS (group A) and from 459 patients with heart failure (HF) without AS (group B) were retrospectively examined to provide an estimation of LVET values in patients with normal and reduced flow status, with and without AS.

The time range of the study was set from February 2016 to September 2019. Exclusion criteria were: presence of mechanical or biological aortic prosthetic valve (52 patients in group A and 39 in group B), significant intraventricular peak pressure gradient ≥ 30 mmHg (2 patients in group A and 9 in group B), unicuspid valve (1 patient in group A), and poor image quality. Finally, group A included 152 patients and group B 378 patients. Demographic, clinical, laboratory and follow-up data were also collected from in-hospital records.

### 2.2. Echocardiographic Examination

Doppler-echocardiographic examinations were performed with a GE Vivid E9, E80 or S6 (GE Health Care, Milwaukee, WI, USA) or Philips EPIQ 7c (Philips, Amsterdam, The Netherlands) ultrasound scanner. Echocardiographic images were stored in a digital format. All measurements were performed by 2 experienced investigators using the ComPACS software Rev. 10.10.20 (Medimatic, Genova, Italy) according with the current European Association of Cardiovascular Imaging (EACVI)/American Society of Echocardiography (ASE) guidelines [7]. LV end-diastolic and end-systolic volumes were calculated using the biplane Simpson method. Right ventricle (RV) function was assessed by the tricuspid annular plane systolic excursion (TAPSE) on the M-mode trace [7]. Grading of valve regurgitation and stenosis was defined following the European Society of Cardiology (ESC) guidelines [1] and the EACVI recommendations [8]. LV diastolic function was assessed according with the algorithm reported in current recommendations [9]. For each Doppler measurement, values were obtained from the average of 3 cardiac cycles in sinus rhythm and 5 cardiac cycles in atrial fibrillation. For AS assessment, AVA was calculated through the continuity equation starting with LVOT diameter measured in parasternal long-axis view, and LVOT time–velocity integral (TVI) on PW Doppler trace and transaortic valve TVI on CW Doppler recording [2]. Doppler recordings were performed in apical 5-chamber and 3-chamber views and, if necessary, in right parasternal view in order to ensure better alignment of the Doppler beam.

### 2.3. AS Severity Definition

AS severity was defined according with current recommendations [1]. Moderate AS (MAS) was defined as AVA between 1 and 1.5 cm^2^ and MG or V_max_ < 40 mmHg and <4 m/s, respectively. In discordant cases in which an AVA < 1 cm^2^ and MG < 40 mmHg or V_max_ < 4 m/s were found along with a low flow (LF) status, pseudosevere AS was differentiated from true LF low-gradient (LG) SAS by dobutamine stress echocardiography, cardiac catheterization (to obtain AVA) or a semi-quantitative assessment of calcium score (eCS) ≥3 [10], and re-classified as MAS. The ratio of continuous wave Doppler acceleration time/ejection time (AT/ET) as an additional marker of SAS was also calculated [11].

### 2.4. LV Systolic Output Measures

The LV stroke distance (SD, cm) was calculated as the TVI of the LVOT flow velocity recorded in the apical 5-chamber or long-axis view (Figure 1). The LVET (ms) was measured as the time interval between the onset and the end of the LVOT flow PW velocity recording (Figure 1). Because the stroke volume (SV) depends on body surface area (BSA), it was indexed by BSA (SVI, mL/m^2^) and used to define a LF status when ≤35 mL/m^2^. The flow rate (FR) (ml/s) was calculated by dividing the SV by the LVET and utilized to define a slow flow (SF) status when ≤210 mL/s [12].

### 2.5. Classification of Flow Status

Patients with and without AS were classified according to their normal or abnormal flow status. The normal flow (NF) status was defined as the combination of normal SVI and FR, whereas the abnormal flow status was defined as the presence of LF or SF or both. Therefore, abnormal flow patients were those with LF but normal FR, SF but normal SVI, or concomitant LF and SF.

### 2.6. Statistical Analysis and Study Endpoints

Normal distribution was tested with the Kolmogorov–Smirnov test. Continuous variables were expressed as mean and standard deviation or median values with 25th and 75th percentiles if normally or non-normally distributed, respectively. Categorical variables were reported as counts and percentages. For continuous normal variables, Student’s t-test and ANOVA were used for comparisons between two or more than two unpaired groups, respectively. For continuous non-normal variables, the Mann–Whitney U test and the Kruskal–Wallis test were used for comparisons between two or more than two unpaired groups, respectively. Categorical variables were compared by the chi-square test. Pearson correlation was used for normally distributed variables, whereas Spearman correlation was used if at least one variable had nonparametric distribution.

The correlations between LVET, flow measures (SVI and FR), HR, AS grade (no-AS, MAS and SAS), LVEF, SBP and mitral regurgitation (MR) were studied by univariate linear regression analysis. The covariates statistically correlated with LVET were included in the multivariate stepwise linear regression model to find the independent determinants of LVET. In the multivariate analysis, R-square for each step and changes in R-square were analyzed. Due to strong collinearity between SVI and FR (R > 0.5), 2 different multivariate models were performed, each one including either SVI or FR. Subgroups of interest for repeated analysis were identified by LVEF (above and below 50%), SVI (above and below 35 mL/m^2^) and/or FR (above and below 210 mL/s). LVETI was derived from linear regression analysis. LVET and LVETI mean values relative to different AS subgroups were compared. The receiver operating characteristic (ROC) curves for the detection of SAS by LVET and LVETI were calculated and compared with a z-test.

The association with surgical or percutaneous aortic valve intervention at follow-up was compared among tertiles of LVETI with the log-rank test and Cox regression analysis. This outcome endpoint was chosen as a “SAS-specific” exploratory endpoint; other endpoints such as mortality or HF hospitalization would not have been specific for SAS in our population, including severe HF patients without AS. The exploratory analysis was repeated in the abnormal flow patient subgroup. Follow-up and vital status check were performed through the informatic medical platform of the local health unit.

Data were analyzed using IBM SPSS Statistics (version 24) (IBM, Armonk, NY, USA) and MedCalc (version 11.2.1.0) (MedCalc Software, Ostend, Belgium). Differences were considered statistically significant at *p* < 0.05.

## 3. Results

### 3.1. Patient Characteristics

Clinical and echocardiographic characteristics of patients are reported in Table 1. Group A included 152 patients with AS, while group B included 378 patients without AS. SAS was found in 79 patients; among those, 30 patients had LF-LG AS, whose severity was confirmed by cardiac catheterization (*n* = 23), dobutamine stress echocardiography (*n* = 3), and calcium assessment (*n* = 4). MAS was present in 73 patients, among which 11 patients had pseudosevere AS correctly diagnosed using cardiac catheterization (*n* = 5), dobutamine stress echocardiography (*n* = 1), and calcium score (*n* = 4). Patients with AS were older, with higher LVEF, SVI and LVET but similar FR to patients without AS (Table 1). Distribution of flow in terms of SVI and FR is shown in Figure 2. Patients of group B had a higher prevalence of LF without SF than patients with SAS (20% vs. 8%, respectively; *p* = 0.029; Figure 2). Among group A patients, 20% had LF without SF, whereas 13% had SF without LF. LVET was significantly longer in AS patients compared to patients without AS (Figure 3), but it failed to discriminate SAS vs MAS in the entire cohort. Indeed, analyzing LVET in different forward flow conditions, it was longer in SAS than in MAS patients with the NF condition, although it was not different between SAS and MAS patients with abnormal flow status (Figure 3).

### 3.2. Determinants of LVET and LVETI Derivation

Correlations and linear regression analysis are shown in Table 2 and Figure 4. On univariate analysis, SVI (Beta = 0.612; *p* < 0.001), AS grade (Beta = 0.383; *p* < 0.001), LVEF (Beta = 0.385; *p* < 0.001) and FR (Beta = 0.11; *p* = 0.011) were directly associated with LVET, whereas HR (Beta = −0.546; *p* < 0.001) and MR (Beta = −0.231; *p* < 0.001) were inversely associated. Since SVI and FR were correlated in overall patients and subgroups (overall patients: R = 0.761, *p* < 0.001; no-AS subgroup: R = 0.734, *p* < 0.001; MAS subgroup: R = 0.871, *p* < 0.001; SAS subgroup: R = 0.876, *p* < 0.001, Appendix A), two multivariate models were performed, either with SVI or FR, to avoid collinearity. On the first multivariate analysis, SVI (Beta = 0.354; *p* < 0.001), HR (Beta = 0.385; *p* < 0.001), AS grade (Beta = 0.301; *p* < 0.001) and LVEF (Beta = 0.108; *p* = 0.001) were independently associated with LVET. On the multivariate analysis including FR, only HR, AS grade and LVEF were associated with LVET (Table 2). Single steps in regression analysis are shown in Table 3. The highest R-square was observed in the model with SVI; therefore, this one was used to derive the indexation formula for LVET. Since the contribution of LVEF was low (change in R-square 0.009 from the model with SVI, HR, and AS grade), LVEF was not included in the final formula (Table 2). The analysis was repeated in the flow subgroups (NF, abnormal flow, LF, SF) and LVEF subgroups (above and below 50%), showing consistently SVI, HR and AS grade as independent determinants of LVET in all subgroups (Table 4).

### 3.3. Analysis with LVETI

When the entire patient cohort was considered, LVETI was significantly higher in patients with AS than in those without, and in patients with SAS than in those with MAS (Figure 5). When the analysis was performed in different flow conditions (NF or abnormal flow status), LVETI was always longer in patients with SAS than in those with MAS or without AS (Figure 5). On ROC analysis, LVETI showed significantly better accuracy in detecting SAS compared to LVET (AUC 0.812 vs. 0.755, *p* = 0.005; Figure 6, upper left panel) and this was observed also in patients with abnormal flow status (AUC 0.800 vs. 0.753, *p* = 0.026; Figure 6, upper right panel).

### 3.4. LVETI and Outcome

After a median follow-up of 2.6 years (interquartile range: 2.0–3.2 years), there were 72 AV interventions (49 with trans-catheter AV replacement and 23 with surgical AV replacement). The median time from echocardiography to AV intervention was 55 days (interquartile range: 15–173 days). Dividing the entire patients’ cohort in LVETI tertiles (<291 ms; 291–322 ms; >322 ms), there were 5 AV interventions in the lowest tertile, 18 in the medium tertile, and 49 in the highest tertile at follow-up (*p* < 0.001). The medium and the highest tertiles of LVETI were associated with AV intervention at follow-up (HR 3.57, 95% confidence interval 1.33–9.63, *p* = 0.012 and HR 10.78, confidence interval 4.29–27.09, *p* < 0.001, respectively) compared to the lowest tertile (Figure 6, lower left panel). In the abnormal flow patient subgroup, the highest tertile of LVETI was associated with AV intervention at follow-up compared to the lowest one (HR 7.33, 95% confidence interval 2.54–21.13, *p* < 0.001) (Figure 6, lower right panel).

## 4. Discussion

In our study, we found that: (1) SVI, HR and AS grade are independent determinants of LVET regardless of forward flow conditions; (2) LVETI derived by indexing LVET for SV and HR discriminates AS grades better than uncorrected LVET, also in different hemodynamic conditions, i.e., in AS patients with normal and abnormal flow status; and (3) the highest tertile of LVETI is associated with the need for AV intervention during follow-up, regardless of flow status.

### 4.1. Pathophysiology of LVET

As recognized by the current echocardiographic guidelines [1], AVA measurement presents limitations in clinical practice; therefore, other variables need to be considered in clinical decision-making for patients with SAS, including MG, flow status, ventricular size and function, wall thickness, presence and degree of AV calcification, blood pressure, and patient functional status. However, all these variables also have limitations, and it is unclear how they interact. Thus, the search for the optimal echocardiographic measurement to quantify AS severity is still going on. LVETI could be a practical solution.

From a pathophysiological point of view, the duration of LVET is influenced by various factors. The shortening of LVET may be determined by heart failure, diminished preload, positive inotropic agents, mitral stenosis and regurgitation [13,14]. Specifically, in heart failure there is a prolongation of the isovolumic contraction time and reduced rate of LV pressure rise (LV dP/dt); therefore, the LVET is shortened, owing to both the delayed onset of ejection and the decreased ability of the heart to maintain high LV pressures during the ejection period, leading to lesser extent of fiber shortening and reduced SV. Conversely, the principal determinant of LVET lengthening is AS, as it determines obstruction to blood flow [13]. According to these observations, LVET was shorter in our patients with heart failure without AS, and increased in AS patients.

### 4.2. LVET in Aortic Stenosis

The relationship between LVET and AVA in AS is not linear. This is evidenced by the Gorlin and Gorlin equation for the calculation of AVA at cardiac catheterization [6]:$$\mathrm{AVA}=\frac{\mathrm{CO}/\left(\mathrm{LVET}\times \mathrm{HR}\right)}{44.3\times \sqrt{\mathrm{MG}}},$$
where AVA increases with increasing cardiac output (CO), while being inversely related with LVET, HR and the square root of the trans-valvular MG. Considering that SV=CO/HR, LVET can be predicted using the following equation:$$\mathrm{LVET}=k\times \frac{\mathrm{SV}}{\mathrm{AVA}\times \sqrt{\mathrm{MG}}}.$$

This equation also evidences the relationship between LVET and SV, showing that the effect of decreasing AVA on LVET lengthening may be counterbalanced by the effect of decreasing SV on LVET shortening [15]. Therefore, if we compare patients with good and poor ventricular function, LVET will be longer in the former due to a higher SV, at any given AVA [16,17]. The joint effects of AVA, SV and HR on LVET have always been considered as a limiting factor to the application of LVET in the assessment of AS severity. This study, for the first time, changed the perspective, disentangling LVETI as a marker of AS.

### 4.3. Assessment of Forward Flow Status

We assessed LV forward flow using SVI, with a 35 mL/m^2^ cutoff value for LF [1,2], and FR, with a 210 mL/s cutoff value for SF [11]. In recent years, SF has been reported to identify the flow reduction that may induce pseudosevere AS [18]. We observed a high correlation between SVI and FR, which increased with worsening AS grades (from R = 0.734 in the no-AS group to R = 0.876 in the SAS group) (Appendix A). LF without SF was significantly less frequent in patients with SAS than in those without AS (8% vs 20%, Figure 2). A decrescendo proportion of LF with normal FR with crescent AS severity was also reported in other studies (Figure 7) [12,19]. The opposite was observed in the prevalence of SF without LF, a condition which was more frequent in patients with SAS compared to those without. This was also observed in other AS study cohorts [11,18] (Figure 7). Both the lower frequency of LF without SF and the higher frequency of SF without LF in SAS patients are related to higher LVET values and support the independent role of AS in prolonging LVET.

### 4.4. LVETI for SAS Identification

In this study, the independent association between LVETI and AS grading was observed in overall patients and in LVEF and flow subgroups, even with LF and SF status. LVETI derived from the analysis of overall patients was compared with LVET in identifying SAS. Contrastingly to LVET, LVETI was longer in SAS than in MAS and patients without AS, and both in NF and in SF or LF subgroups. On the ROC analysis, this translated to a better accuracy of LVETI than LVET in detecting SAS, even in the abnormal flow subgroups (Figure 6). These data clearly indicate the potential role of LVETI as a complementary evaluation in the AS severity assessment, especially in controversial or borderline cases.

### 4.5. LVETI and Outcome

In this study, we provided exploratory data about the association between LVETI and outcome. The tertile of patients with longest LVETI frequently underwent AV interventions at follow-up (68%). Even in the abnormal flow patient subgroup, the longest LVETI tertile showed a strong association with AV interventions at follow-up (Figure 6). Since time to AV intervention was short (median time 55 days) we infer that all these data corroborate LVETI as a SAS measure.

### 4.6. Study Limitations and Perspectives

(1) This is a retrospective investigation. However, because our digital platform allows comprehensive storage of echocardiographic examinations, all necessary images were available for the required measurements;

(2) LVET indexation requires an estimate of SV and LVOT diameter, which can be measured only in the presence of a parasternal long-axis view with sufficient image quality. However, if the LVOT diameter cannot be obtained, other methods to estimate SV may be considered (e.g., difference between end-diastolic and end-systolic volumes, in the absence of significant MR);

(3) The LVETI formula derived from the entire patient cohort was applied also to the abnormal flow subgroups. We recognize that, in these subgroups, SVI and HR may account differently with respect to overall patients (Table 4), and thus specific formulas might be necessary for the best accuracy in identifying SAS. However, although our data allowed the derivation of such formulas (Appendix A), we did not apply them in our study, because this investigation was not designed for this purpose. Future investigations are indicated to explore the value of different LVETI formulas in specific AS hemodynamic subgroups;

(4) The use of AV intervention as an outcome measure for SAS is limited because some SAS patients may not have undergone intervention, for example due to comorbidities. Moreover, it was the only outcome measure utilized in this study. We recognize that dedicated, prospective outcome studies are needed, possibly based on harder end-points (such as survival);

(5) Namasivayam et al. showed the lack of prognostic value of AVA ≤ 1 cm^2^ in SF-LG AS (FR ≤ 210 mL/s) [12]. Thus, the contribution of LVETI in this subset of patients could be of particular value and should be addressed in appropriate investigations;

(6) In our data, LVET seemed not to be conditioned by SBP at admission (Table 2) but this observation needs confirmation using SBP at the moment of echocardiography. If this will be confirmed, LVETI might also be proposed for hypertensive patients with high valvulo-arterial impedance, for whom current guidelines suggest repeating echocardiography after the normalization of blood pressure [1];

(7) Finally, this was a single-center investigation with a limited number of patients with SAS. A large number of these patients could be included in a multicenter study.

## 5. Conclusions

In this study, LVET was corrected by its determinants to derive LVETI, a new index of AS severity, which may be helpful in recognizing SAS, especially in controversial cases. Future, large-scale validation studies of LVETI as a predictor of AS severity are needed.

## Figures and Tables

**Figure 1 jcm-11-01877-f001:**
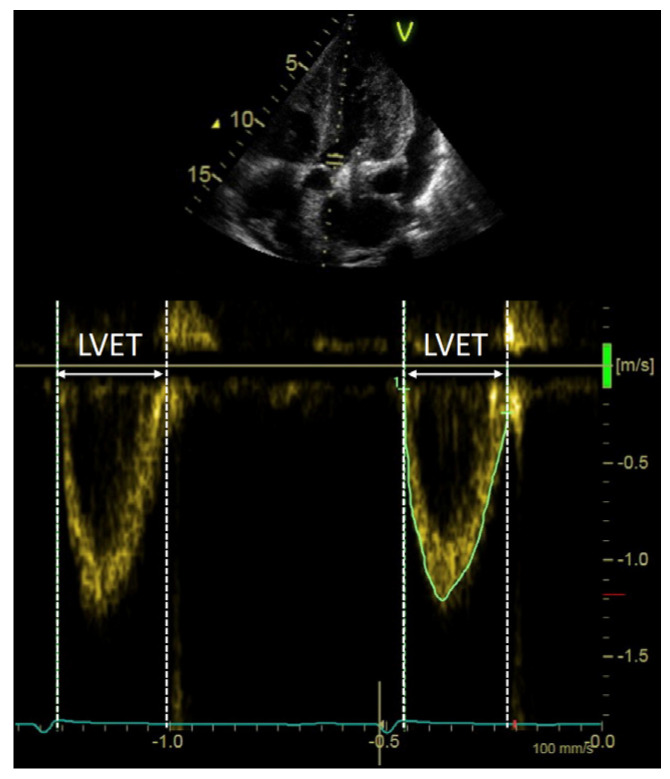
Left ventricular outflow tract velocity profile showing measurement of left ventricular ejection time (LVET).

**Figure 2 jcm-11-01877-f002:**
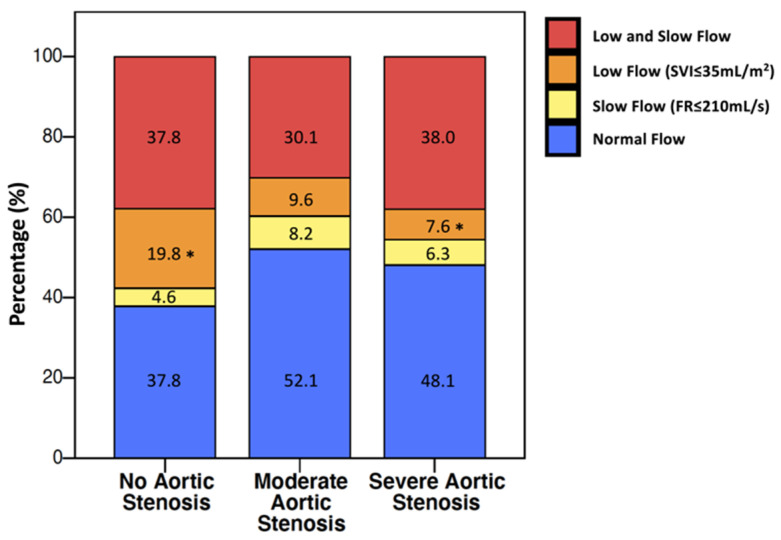
Distribution of flow status profiles according to stroke volume index (SVI) and flow rate (FR). * Adjusted for multiple comparisons *p* value = 0.029. All the other comparisons were not significant.

**Figure 3 jcm-11-01877-f003:**
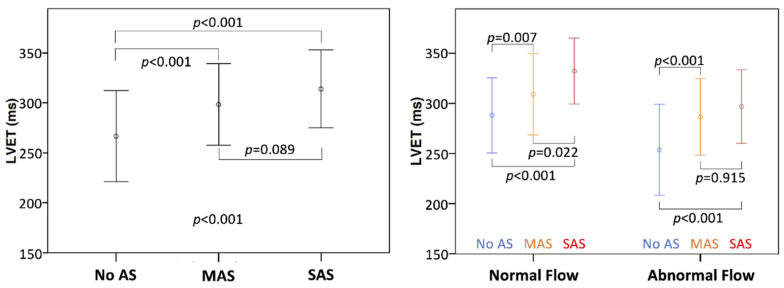
Mean left ventricular ejection time (LVET) according to aortic stenosis (AS) grading (**left**) and to flow status and AS grading (**right**). Abnormal flow is identified by stroke volume index ≤ 35 mL/m^2^ or flow rate ≤ 210 mL/s. MAS, moderate AS; SAS, severe AS.

**Figure 4 jcm-11-01877-f004:**
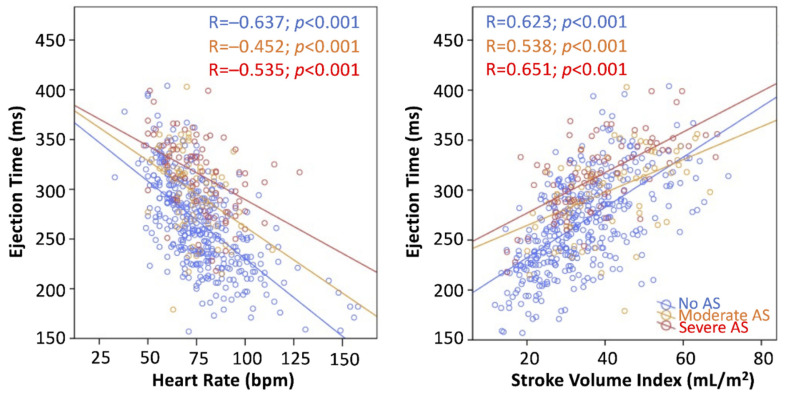
Scatterplot of left ventricular ejection time versus heart rate (**left**) and stroke volume index (**right**), according to aortic stenosis (AS) grading.

**Figure 5 jcm-11-01877-f005:**
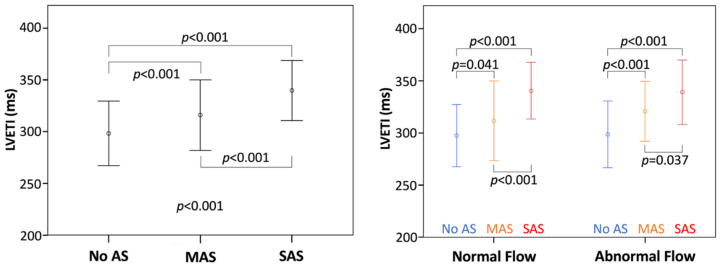
Mean left ventricular ejection time index (LVETI) according to aortic stenosis (AS) grading (**left**) and to flow status and AS grading (**right**). Abnormal flow is identified by stroke volume index ≤ 35 mL/m^2^ or flow rate ≤ 210 mL/s. MAS, moderate AS; SAS, severe AS.

**Figure 6 jcm-11-01877-f006:**
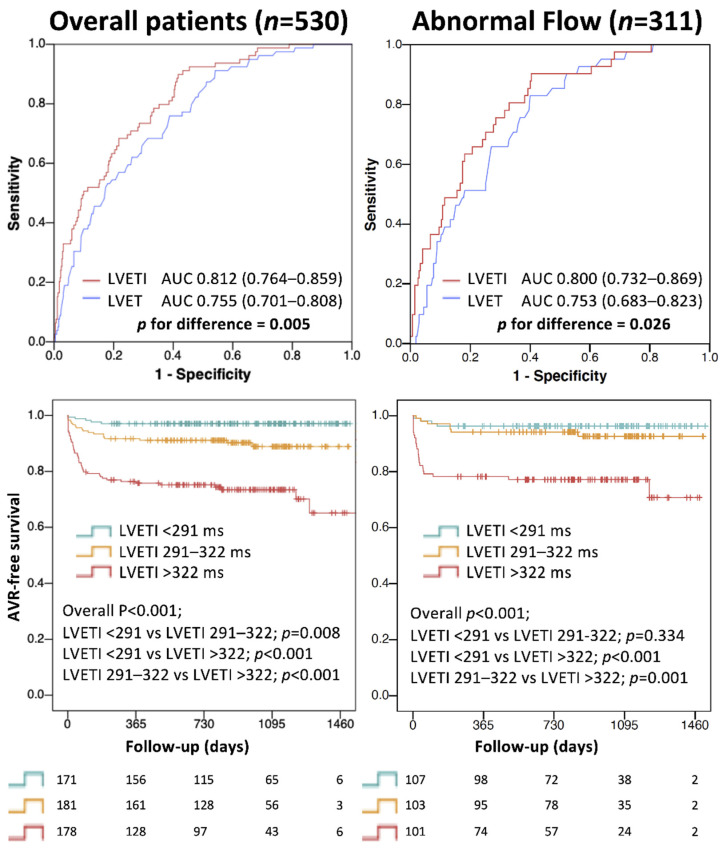
Upper: receiver operating characteristic curves representing the accuracy of left ventricular ejection time (LVET) and LVET index (LVETI) in detecting severe aortic stenosis in overall patients (**upper left**) and in the abnormal flow subgroup (**upper right**). Lower: freedom from aortic valve replacement (AVR) according to tertiles of LVETI in overall patients (**lower left**) and in the abnormal flow subgroup (**lower right**). AUC, area under the curve.

**Figure 7 jcm-11-01877-f007:**
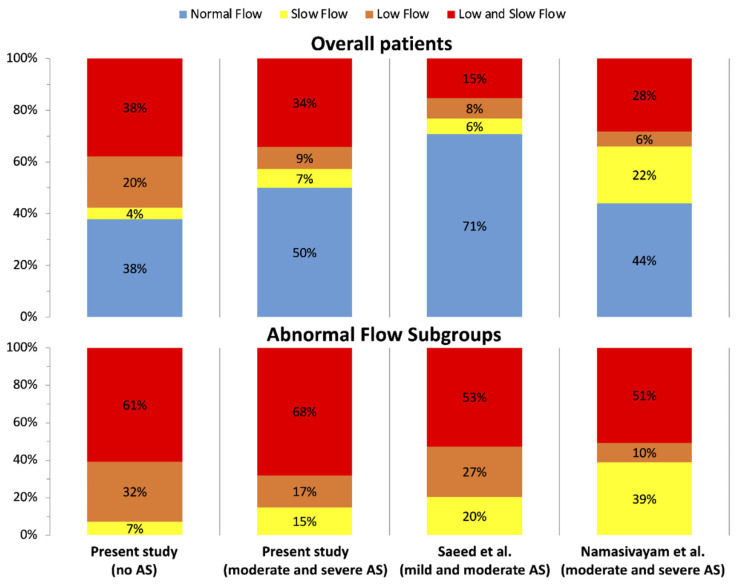
Distribution of flow in our cohort and previous studies from Saeed et al. [19] and Namasivayam et al. [12] on various AS grade patients.

**Table 1 jcm-11-01877-t001:** Patient characteristics.

	Overall Patients *n* = 530	AS (Group A) *n* = 152	Moderate AS *n* = 73	Severe AS *n* = 79	No-AS (Group B) *n* = 378	*p* *	*p* **
Age (years)	80 (72–86)	82 (77–86)	83 (77–87)	81 (76–85)	78 (70–85)	<0.001	0.001
Male (*n*)	270 (51)	77 (51)	35 (48)	42 (53)	193 (51)	0.934	0.810
BSA (m^2^)	1.85 (1.7–1.99)	1.83 (1.71–1.97)	1.83 (1.71–1.97)	1.85 (1.71–1.96)	1.86 (1.69–2)	0.364	0.468
HR (bpm)	74 (64–85)	73 (65–85)	71 (66–80)	76 (64–90)	75 (64–85)	0.807	0.216
SBP (mmHg)	140 (120–160)	130 (120–150)	130 (120–150)	130 (115–145)	140 (120–160)	0.005	0.011
History of HF (*n*)	154 (29)	61 (40)	24 (33)	37 (47)	93 (25)	<0.001	<0.001
AF (*n*)	223 (42)	67 (44)	32 (44)	35 (44)	156 (41)	0.554	0.838
Hypertension (*n*)	397 (75)	121 (80)	56 (77)	65 (82)	276 (73)	0.114	0.209
DM (*n*)	151 (29)	46 (30)	22 (30)	24 (30)	105 (28)	0.566	0.848
CKD (*n*)	161 (30)	37 (24)	18 (25)	19 (24)	124 (33)	0.055	0.159
CAD (*n*)	182 (34)	65 (43)	27 (38)	38 (48)	117 (31)	0.008	0.012
Loop diuretic (*n*)	454 (86)	120 (79)	59 (81)	61 (77)	334 (88)	0.001	0.003
Beta-blockers (*n*)	357 (67)	83 (54)	40 (55)	43 (54)	274 (73)	<0.001	<0.001
Renin-angiotensin system blockers	366 (69)	111 (73)	53 (73)	58 (73)	255 (68)	0.210	0.453
Mineralcorticoid receptor antagonist	215 (41)	53 (35)	29 (40)	24 (30)	162 (43)	0.076	0.089
LVEDV (mL)	113 (88–145)	115 (90–141)	114 (83–135)	117 (94–145)	111 (86–145)	0.713	0.493
LVESV (mL)	55 (36–91)	52 (37–78)	49 (36–72)	53 (37–89)	56 (36–98)	0.211	0.389
LVEF (%)	50 (34–59)	55 (39–61)	53 (39–60)	56 (38–62)	48 (32–58)	0.002	0.007
SVI (mL/m^2^)	34 (27–42)	37 (31–49)	38 (31–49)	37 (29–48)	32 (26–40)	<0.001	<0.001
Low Flow (*n*)	283 (53)	65 (43)	29 (40)	36 (46)	218 (58)	0.002	0.006
FR (mL/s)	224 (187–274)	226 (190–276)	233 (200–293)	222 (178–261)	223 (186–268)	0.628	0.23
Slow Flow (*n*)	223 (42)	63 (41)	28 (38)	35 (44)	160 (42)	0.853	0.746
LVET (ms)	278 (48)	306 (40)	298 (41)	314 (39)	267 (46)	<0.001	<0.001
LAVI (mL/m^2^)	49 (40–58)	50 (43–63)	53 (44–65)	49 (41–62)	49 (39–57)	0.013	0.013
MR (*n*)	251 (47)	56 (37)	33 (45)	23 (29)	195 (52)	0.002	0.001
SPAP (mmHg)	43 (35–53)	38 (30–50)	40 (31–50)	37 (30–47)	45 (38–53)	<0.001	<0.001
AT/ET	-	0.35 (0.31–0.38)	0.33 (0.29–0.36)	0.37 (0.34–0.41)	-	-	<0.001
AVA (cm^2^)	-	0.9 (0.7–1.1)	1.2 (1–1.5)	0.7 (0.6–0.9)	-	-	<0.001
Mean Gradient (mmHg)	-	29 (21–44)	21 (16–26)	43 (30–56)	-	-	<0.001
V Max (m/s)	-	3.7 (3.2–4.6)	3.2 (3–3.4)	4.3 (3.6–4.8)	-	-	<0.001

Continuous non-parametric variables are expressed as median (25th and 75th percentiles), parametric variables as mean (standard deviation) and categorical variables as counts (frequency percentages). AF, atrial fibrillation; AS, aortic stenosis; AT, continuous wave Doppler acceleration time; AVA, aortic valve area; BSA, body surface area; CAD, coronary artery disease; CKD, chronic kidney disease; DM, diabetes mellitus; ET, continuous wave Doppler ejection time; FR, flow rate; HF, heart failure; HR, heart rate; LAVI, left atrial volume index; LVEDV, left ventricle end-diastolic volume; LVEF, left ventricle ejection fraction; LVESV, left ventricle end-systolic volume; LVET, left ventricle ejection time; MR, mitral regurgitation; SBP, systolic blood pressure; SPAP, systolic pulmonary artery pressure; SVI, stroke volume index; V Max, maximal aortic valve velocity. * Between no-AS and AS. ** Among no-AS, MAS and SAS.

**Table 2 jcm-11-01877-t002:** Determinants of LVET in overall patients: correlations and linear regression analysis.

	R	*p*	Univariate Beta	*p*	Multivariate Beta Model with SVI	*p*	Multivariate Beta Model with FR	*p*
SVI	0.618	<0.001	0.612	<0.001	0.354	<0.001	Not tested	
HR	−0.539	<0.001	−0.546	<0.001	−0.385	<0.001	−0.497	<0.001
AS grade	0.394	<0.001	0.383	<0.001	0.301	<0.001	0.349	<0.001
LVEF	0.372	<0.001	0.385	<0.001	0.108	0.001	0.247	<0.001
SBP		0.4	0.052	0.254				
MR	−0.226	<0.001	−0.231	<0.001		0.427		0.166
FR	0.145	0.001	0.11	0.011	Not tested			0.42
Regression equation for indexing LVET LVETI = LVET – (1.452 × SVI) + (1.05 × HR)								

**Table 3 jcm-11-01877-t003:** Incremental models with stepwise linear regression analysis.

	Global R-Square	Change in R-Square from Previous Model	*p* for Difference from Previous Model
Model with SVI			
SVI	0.374	0.374	<0.001
SVI + HR	0.485	0.111	<0.001
SVI + HR + AS grade	0.575	0.090	<0.001
SVI + HR + AS grade + LVEF	0.584	0.009	0.001
Model with FR			
HR	0.298	0.298	<0.001
HR + AS grade	0.442	0.144	<0.001
HR + AS grade + LVEF	0.500	0.058	<0.001

**Table 4 jcm-11-01877-t004:** Flow and LVEF subgroup analysis.

	R	*p*	Uni Beta	*p*	Multi Beta	*p*
Low Flow (SVI ≤ 35 mL/m^2^) (*n* = 283)						
SVI	0.52	<0.001	0.539	<0.001	0.304	<0.001
HR	−0.504	<0.001	−0.526	<0.001	−0.396	<0.001
LVEF	0.25	<0.001	0.236	<0.001	0.123	0.005
AS grade	0.388	<0.001	0.389	<0.001	0.342	<0.001
SBP		0.759	0.007	0.905		
MR	−0.144	0.015	−0.137	0.021		0.478
Slow Flow (FR ≤ 210 mL/s) (*n* = 223)						
SVI	0.766	<0.001	0.781	<0.001	0.607	<0.001
HR	−0.563	<0.001	−0.575	<0.001	−0.288	<0.001
LVEF	0.394	<0.001	0.403	<0.001		0.061
AS grade	0.358	<0.001	0.327	<0.001	0.222	<0.001
SBP		0.523	0.073	0.3		
MR	−0.2	0.003	−0.195	0.004		0.566
Normal Flow (*n* = 219)						
SVI	0.462	<0.001	0.436	<0.001	0.265	<0.001
HR	−0.403	<0.001	−0.388	<0.001	−0.367	<0.001
LVEF	0.242	<0.001	0.211	0.002		0.74
AS grade	0.432	<0.001	0.417	<0.001	0.399	<0.001
SBP		0.613	−0.053	0.454		
MR		0.051	−0.162	0.016		0.129
Abnormal Flow (*n* = 311)						
SVI	0.611	<0.001	0.66	<0.001	0.423	<0.001
HR	−0.549	<0.001	−0.562	<0.001	−0.359	<0.001
LVEF	0.314	<0.001	0.321	<0.001	0.113	0.004
AS grade	0.379	<0.001	0.353	<0.001	0.279	<0.001
SBP		0.889	0.043	0.469		
MR	−0.185	0.001	−0.179	0.001		0.611
LVEF 50% (*n* = 263)						
SVI	0.567	<0.001	0.553	<0.001	0.352	<0.001
HR	−0.471	<0.001	−0.492	<0.001	−0.382	<0.001
LVEF		0.158	0.047	0.451		
AS grade	0.417	<0.001	0.401	<0.001	0.314	<0.001
SBP		0.987	−0.019	0.766		
MR		0.097	−0.105	0.090		
LVEF < 50% (*n* = 267)						
SVI	0.571	<0.001	0.548	<0.001	0.329	<0.001
HR	−0.536	<0.001	−0.539	<0.001	−0.431	<0.001
LVEF	0.282	<0.001	0.277	<0.001	0.15	0.001
AS grade	0.339	<0.001	0.321	<0.001	0.332	<0.001
SBP		0.677	0.081	0.204		
MR	−0.135	0.028	−0.139	0.023		0.422

LVETI, left ventricle ejection time index; other abbreviations as in Table 1.

## Data Availability

Data available on justified request.

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
