# Peer review of "Value of Left Ventricular Indexed Ejection Time to Characterize the Severity of Aortic Stenosis"

_jcm, 2022, doi:10.3390/jcm11071877_

Round 1

Reviewer 1 Report

The authors proposed verify if correcting LVET (LVET index, LVETI) by its determinants is helpful for the assessment of AS severity irrespectively of hemodynamic conditions.

The authors analyzed retrospectively studied 152 patients with AS and 378 patients with heart failure and no AS.

In the multivariate model, the authors stated LVET (assessed with pulsed-wave Doppler) showed a strong correlation with stroke volume index (SVI) (Beta 0.354; P<0.001), HR (-0.385; P<0.001), AS grade (Beta 0.301; P<0.001) and, less significantly, ejection fraction (LVEF) (Beta 0.108; P=0.001).

The authors concluded LVETI correlatd with AS severity better than uncorrected LVET, independently from the hemodynamic conditions, and may help to discriminate severe AS.

Left ventricular ejection time (LVET) is simply defined as the time interval from aortic valve opening to aortic valve closure, and is the phase of systole during which the left ventricle ejects blood into the aorta. Despite this simple definition, it is important to notice that methods of measuring the cardiac time intervals including LVET have changed markedly over the last 50 years.

This work is nevertheless interesting for an 'old' marker that seeks to provide arguments for a current utility.

Major revision:

  1. The authors seem not to take into account the latest news on the subject. The Acceleration time was not evaluated in this study. AT (acceleration time) and ET (ejection time = LVET) are usually used in aortic stenosis evaluation. The combination of the two gives an idea about the severity of AR and especially its prognosis by combining the severity of the disease and being a marker of heart failure. (J Am Heart Assoc. 2021;10:e021873. DOI: 10.1161/JAHA.121.021873) By calculating their own ET index for each patient, the authors could have avoided the misclassification of heart rate and flow.
  2. The authors did not report the patients' medications. However, it was been shown that there is a strong link between the therapies and LVET (doi:10.1002/ejhf.2125).
  3. The study was single-centre with low statistical power, including less than 80 patients with severe aortic stenosis.
  4. At the methodological level, the authors approached the notion of prediction with the calculation of AUC while there was no derivation data set or validation data set. The metholdogy used did not support a conclusion related to prediction.
  5. The authors based their first model with the LVET as the outcome of the model. Then they focused on prediction with another outcome, the occurrence of severe aortic stenosis. (SAS) The prediction models were not detailed and we could not appreciate the construction of the model in state. It might have been necessary to build first logistic regression models with SAS as an outcome and to limit the performance of this model by C-Statistics presentation, without addressing the notion of prediction which seems to be methodologically wrong.
  6. The authors stated that the LVET was highly correlated with heart rate or stroke index. We could note a correlation. In view of the correlation coefficient and the available data, this correlation was rather intermediate than strong.

Author Response

  1. The authors seem not to take into account the latest news on the subject. The Acceleration time was not evaluated in this study. AT (acceleration time) and ET (ejection time = LVET) are usually used in aortic stenosis evaluation. The combination of the two gives an idea about the severity of AR and especially its prognosis by combining the severity of the disease and being a marker of heart failure. (J Am Heart Assoc. 2021;10:e021873. DOI: 10.1161/JAHA.121.021873) By calculating their own ET index for each patient, the authors could have avoided the misclassification of heart rate and flow.

We thank the Reviewer for giving us the opportunity to include the AT/ET ratio in our study. Values of the AT/ET ratio are now reported in table 1. Patients with severe aortic stenosis had a ratio >0.35. The paper indicated by the Reviewer is now cited as reference 11.

  1. The authors did not report the patients' medications. However, it was been shown that there is a strong link between the therapies and LVET (doi:10.1002/ejhf.2125).

We thank the Reviewer for this suggestion. Patients’ medications are now reported in table 1.

  1. The study was single-centre with low statistical power, including less than 80 patients with severe aortic stenosis.

We thank the Reviewer for the opportunity to clarify this aspect, which has now been addressed in the Limitations section (point 7): “Finally, this is a single-center investigation with a limited number of patients with severe aortic stenosis. A large number of these patients could be included in a multicenter study.”

  1. At the methodological level, the authors approached the notion of prediction with the calculation of AUC while there was no derivation data set or validation data set. The metholdogy used did not support a conclusion related to prediction.

We thank the Reviewer for this comment. We agree that the AUC analysis is not a predictive analysis. Thus, in the current version of the manuscript we better described the statistical approach and commented our data to avoid the notion of prediction.

  1. The authors based their first model with the LVET as the outcome of the model. Then they focused on prediction with another outcome, the occurrence of severe aortic stenosis. (SAS) The prediction models were not detailed and we could not appreciate the construction of the model in state. It might have been necessary to build first logistic regression models with SAS as an outcome and to limit the performance of this model by C-Statistics presentation, without addressing the notion of prediction which seems to be methodologically wrong.

As indicated in the answer to previous point 4, the notion of prediction is now eliminated from the current version of the paper. We also realized that description of the study end-points was not clear and this probably generated confusion about the statistical study model. Therefore, we have now also clarified the study end-points.

  1. The authors stated that the LVET was highly correlated with heart rate or stroke index. We could note a correlation. In view of the correlation coefficient and the available data, this correlation was rather intermediate than strong.

According with the Reviewer’s comment, we removed the word “highly”.

We thank the Reviewer very much for his/her helpful insights which allowed us to improve substantially our work.

Reviewer 2 Report

The submitted manuscript titled “VALUE OF LEFT VENTRICULAR INDEXED EJECTION TIME 2 TO CHARACTERIZE SEVERITY OF AORTIC STENOSIS” investigated the relationship between hemodynamic status and ejection time and the clinical value of ejection time in patients with AS. While the focus of this study is interesting and useful in clinical practice, I would like to highlight the following key concerns that need to be addressed to improve the quality of this article.

  1. Add a Figure of the actual measured ejection time.
  2. Why is the current study comparing patients with aortic stenosis and heart failure? Is it better to compare with healthy controls?
  3. In the present study, aortic valve replacement was the only clinical outcome, and selection and confounding bias cannot be ruled out.
  4. It is necessary to examine the significance of ejection time in addition to conventional indices such as peak velocity and mean pressure gradient.
  5. In this study, both stroke volume and flow rate were used for flow, but it is easier to understand if either of them is used.

Author Response

  1. Add a Figure of the actual measured ejection time.

A figure illustrating how to measure ejection time has been added (current figure 1).

  1. Why is the current study comparing patients with aortic stenosis and heart failure? Is it better to compare with healthy controls?

We thank the Reviewer for this question. Patients with heart failure were chosen to provide an estimation of LVET values also in patients with reduced flow status, with and without AS. This is now explained in the Methods section, paragraph 2.1 (Patient groups).

  1. In the present study, aortic valve replacement was the only clinical outcome, and selection and confounding bias cannot be ruled out.

We agree with the Reviewer. We have now better clarified in the Limitations section (point 4) the limitations of the outcome analysis, which was only exploratory in this study.

  1. It is necessary to examine the significance of ejection time in addition to conventional indices such as peak velocity and mean pressure gradient.

Conventional indices of aortic stenosis are not always concordant in defining severity of this disease. Thus, there is a potential for LVETI to be a complementary evaluation in the AS severity assessment, especially in controversial or borderline cases. This is now reported in the discussion section (paragraph 4.4). In the limitation section (point 5) we also addressed the potential contribution of LVETI in the subset of patients in the low flow low gradient AS with flow rate <= 210 ml/s.

  1. In this study, both stroke volume and flow rate were used for flow, but it is easier to understand if either of them is used.

We agree with the Reviewer that use of only stroke volume or flow rate would make our work easier to understand. However, studies from the literature (Namasivayam et al, JACC 2020, ref. #12) suggest that the use of only one of these two measures makes evaluation of flow status in patients with aortic stenosis incomplete. This is why we included both stroke volume and flow rate in our study. We hope that the Reviewer would appreciate our effort to precisely identify flow status using the combination of stroke volume and flow rate evaluation.

We thank the Reviewer very much for his/her helpful insights which allowed us to improve substantially our work.

Round 2

Reviewer 2 Report

Thank you for your response. I have no further comments.